# Nanobodies: site-specific labeling for super-resolution imaging, rapid epitope-mapping and native protein complex isolation

Tino Pleiner[1], Mark Bates[2], Sergei Trakhanov[1], Chung-Tien Lee[3,4], Jan Erik Schliep[5], Hema Chug[1], Marc Böhning[1], Holger Stark[5], Henning Urlaub[3,4], Dirk Görlich[1]*

[1]Department of Cellular Logistics, Max Planck Institute for Biophysical Chemistry, Göttingen, Germany; [2]Department of NanoBiophotonics, Max Planck Institute for Biophysical Chemistry, Göttingen, Germany; [3]Bioanalytical Mass Spectrometry, Max Planck Institute for Biophysical Chemistry, Göttingen, Germany; [4]Bioanalytics, Institute for Clinical Chemistry, University Medical Center Göttingen, Göttingen, Germany; [5]3D Electron Cryo-Microscopy Group, Max Planck Institute for Biophysical Chemistry, Göttingen, Germany

**Abstract** Nanobodies are single-domain antibodies of camelid origin. We generated nanobodies against the vertebrate nuclear pore complex (NPC) and used them in STORM imaging to locate individual NPC proteins with <2 nm epitope-label displacement. For this, we introduced cysteines at specific positions in the nanobody sequence and labeled the resulting proteins with fluorophore-maleimides. As nanobodies are normally stabilized by disulfide-bonded cysteines, this appears counterintuitive. Yet, our analysis showed that this caused no folding problems. Compared to traditional NHS ester-labeling of lysines, the cysteine-maleimide strategy resulted in far less background in fluorescence imaging, it better preserved epitope recognition and it is site-specific. We also devised a rapid epitope-mapping strategy, which relies on crosslinking mass spectrometry and the introduced ectopic cysteines. Finally, we used different anti-nucleoporin nanobodies to purify the major NPC building blocks – each in a single step, with native elution and, as demonstrated, in excellent quality for structural analysis by electron microscopy. The presented strategies are applicable to any nanobody and nanobody-target.

*For correspondence: goerlich@mpibpc.mpg.de

**Competing interests:** The authors declare that no competing interests exist.

## Introduction

Nanobodies represent antigen-binding domains of 'heavy-chain-only' camelid antibodies and are typically selected by phage display from an immune library (*Hamers-Casterman et al., 1993*; *Arbabi Ghahroudi et al., 1997*; *Muyldermans, 2013*). Their small size (~13 kDa), monoclonal nature and high specificity are ideal for applications like affinity purification or protein detection and localization (*Helma et al., 2015*). Their utility as crystallization chaperones is also widely appreciated (*Pardon et al., 2014*; *Desmyter et al., 2015*).

Nanobodies are commonly expressed in *Escherichia coli* and secreted into the oxidative periplasm, where their conserved internal disulfide bond can be formed (*Vincke et al., 2012*; *Pardon et al., 2014*; *Fridy et al., 2014*). Periplasmic expression comes, however, with several drawbacks. For example, it often results in low final yield (*Baneyx and Mujacic, 2004*), probably due to saturation of the secretion machinery and aggregation of precursor proteins in the cytoplasm. The

**eLife digest** Antibodies not only protect humans and other animals against disease-causing bacteria and viruses. They can also be used as tools for medical diagnostics and basic research. Conventional antibodies consist of light and heavy protein chains, and both are required to bind to target molecules (or antigens). Alpacas, llamas and camels, however, possess simpler antibodies that lack light chains and bind to antigens via a single protein domain. Such domains can be produced in "re-programmed" bacteria and are then called nanobodies. Compared to normal antibodies, nanobodies are 10-fold smaller, which is of great advantage in virtually all practical applications.

Pleiner et al. made nanobodies against the nuclear pore complex (or NPC for short) – a nanoscopic machine for transporting large biological molecules in and out of the cell's nucleus. These nanobodies can be linked to dyes called fluorophores and then used to stain NPCs so that they can be observed under a microscope.

When fluorophores were attached, in the traditional way, via the amino acid lysine, all tested nanobodies performed poorly in fluorescence microscopy - pointing to a systematic problem. Pleiner et al. therefore explored an alternative, namely to label nanobodies via engineered cysteines. This was counterintuitive, because nanobodies contain already two other cysteines that must not be modified and that normally form a stabilizing "disulfide" bond. Pleiner et al. found, however, that the labeling reaction is absolutely specific for the engineered surface cysteines when it is performed at low temperature. This strategy consistently yielded imaging reagents that could effectively deliver fluorophores as close as 1-2 nanometers to their antigens. Nanobodies labeled in this way are therefore ideal to exploit the full potential of super-resolution microscopy.

The engineered surface cysteines proved also useful as "position sensors" to report which region of an antigen is actually contacted by a given nanobody.

Nanobodies are also used to purify protein complexes from crude cell extracts by a method called affinity chromatography. Previously, nanobodies were chemically attached to an insoluble matrix, and bound protein complexes were released under conditions that destroy interactions between proteins. Pleiner et al. now replaced the destructive step with a step that uses an enzyme to cut a bond and gently detach the nanobody (along with any bound protein complex) from the matrix. Bound protein complexes thus stay intact and can be studied further. In the future, this strategy can be applied to nanobodies that recognize tags commonly added to proteins (i.e. GFP) to isolate virtually any protein complex for functional assays or structural analyses.

limited set of chaperones and high proteolytic activity in the periplasm also restrict the choices of fusion tags that can be used (*de Marco, 2009*; *Feilmeier et al., 2000*). Furthermore, the purification of periplasmic proteins involves considerably more hands-on time than purification from the cytoplasm.

In typical affinity chromatography applications, nanobodies are covalently attached to a resin, which later necessitates harsh conditions for the elution of bound target proteins (*Rothbauer et al., 2008*; *Fridy et al., 2014*). This is appropriate for an identification, but hardly for any further downstream structural or functional analysis of the purified target proteins. As a workaround, a native isolation of protein A-tagged protein complexes using a specific nanobody modified with a dithiothreitol (DTT)-cleavable crosslinker was recently reported (*Fridy et al., 2015*). However, the achievable yield was modest, as most of the isolated complexes resisted release. Furthermore, the presence of any thiol-reducing agent (like DTT or glutathion) during binding is incompatible with this method.

In traditional indirect immunofluorescence, epitopes are initially decorated with a primary antibody and detected with a fluorophore-labeled secondary one, each around 12–15 nm in size (*Harris et al., 1998*). The effective displacement between label and epitope can reach up to 24–30 nm and thus significantly deteriorate the achievable precision and accuracy of protein localization by super-resolution fluorescence microscopy (*Hell, 2009*; *Huang et al., 2009*). Nanobodies (diameter: 4 nm) are an ideal solution to this problem (*Ries et al., 2012*; *Szymborska et al., 2013*). This, however, requires a direct nanobody labeling. Ideally, labeling should be site-specific, so that the

remaining small displacement between epitope and fluorescent dye can be predicted and corrected for in the measurements.

So far, nanobodies were labeled at lysines by N-hydroxysuccinimide (NHS) ester fluorophores (*Ries et al., 2012*; *Fridy et al., 2014*), which is random and rarely quantitative. As we show below, it also deteriorates signal-to-background ratios or even completely abolishes epitope recognition. A workaround to this basic problem was the addition of a C-terminal oligo-lysine stretch to divert labeling from nanobody framework lysine residues (described for the anti-GFP nanobody 'Enhancer' in *Platonova et al., 2015*). This, however, increases the epitope-label distance again. Furthermore, fluorescent labeling of nanobodies using Sortase A was presented (*Witte et al., 2012*). This strategy is limited to the N- or C-terminus and uses modified fluorophores that are not readily available. Adding an extra C-terminal cysteine (for subsequent maleimide modification) to a periplasmically expressed nanobody was also not a satisfying solution, because it led to a severe reduction in yield and caused extensive dimerization (*Massa et al., 2014*).

Hence, we explored solutions to the above-described limitations of the current nanobody technology. We demonstrate functional cytoplasmic expression of nanobodies with protease-cleavable tags for native affinity purification and with engineered cysteines for site-specific fluorescent labeling. We chose the *Xenopus* nuclear pore complex (NPC) as a model target and developed a toolbox of high-affinity nanobodies against its major constituent proteins, nucleoporins (Nups), which occur in large subcomplexes. Using specific nanobodies, we purified their target protein complexes from *Xenopus* egg extract in a single step with native elution based on proteolytic matrix-release. This allowed a direct analysis of nanobody-purified endogenous Nup complexes by negative stain electron microscopy. Labeling these anti-Nup nanobodies with NHS ester fluorescent dyes for imaging often produced non-functional reagent or significant background staining. We therefore implemented a simple and generally applicable strategy for obtaining site-specifically fluorophore-labeled nanobodies of superior imaging quality. It involves engineered cysteines at the nanobody surface, their modification with maleimide fluorophores, and leaves the internal framework cysteines fully intact. This strategy allowed super-resolution imaging of NPCs with a negligible label displacement and very low background. A novel strategy for rapid mapping of conformational nanobody epitopes via crosslinking mass spectrometry involving the engineered surface cysteines is also presented here.

## Results

NPCs are gateways for nucleocytoplasmic transport. Their very large size of ≈110 MDa not only places them amongst the largest molecular machines, but also poses formidable challenges for any structural and functional investigation (reviewed in *Hurt and Beck, 2015*). Nucleoporins are organized in multiple subcomplexes around a central eightfold rotational symmetry axis. Certainly, the most characteristic subcomplex is the Nup107-Nup160 or Y-shaped complex (*Siniossoglou et al., 2000*; *Vasu et al., 2001*). Its essential role in NPC assembly (*Harel et al., 2003*; *Walther et al., 2003*) as well as its structural organization and relative position within the NPC have been studied intensely (*Bui et al., 2013*; *Eibauer et al., 2015*; *Kelley et al., 2015*; *Stuwe et al., 2015b*; *von Appen et al., 2015*). The inner ring of the NPC scaffold comprises also essential Nup93-containing subcomplexes (*Sachdev et al., 2012*; *Vollmer and Antonin, 2014*), whose stoichiometry and orientation are less understood. Nup93 further anchors the trimeric Nup62•Nup58•Nup54 complex in the central transport channel (*Finlay et al., 1991*; *Hu et al., 1996*; *Chug et al., 2015*; *Stuwe et al., 2015a*).

### Functional cytoplasmic expression of anti-NPC nanobodies

In order to provide new tools for studying NPCs, we generated nanobodies against constituents of the *Xenopus* NPC scaffold, namely Nup85, Nup93, and Nup155, as well as against Nup98 and the Nup62•Nup58•Nup54 complex. The latter two species were included because their Phe-Gly (FG)-repeat domains form a permeability barrier within the central NPC channel (*Hülsmann et al., 2012*). High-affinity nanobodies against all these targets were readily obtained from alpaca immune libraries by phage display.

We noticed that these nanobodies could be produced in the cytoplasm of various *E. coli* strains, as recently reported for other nanobodies (*Olichon and Surrey, 2007*; *Zarschler et al., 2013*; *Djender et al., 2014*). We also observed that fusing nanobodies behind a $His_{14}$-bdNEDD8 module (*Frey and Görlich, 2014*) increased their yield dramatically (*Figure 1a*).

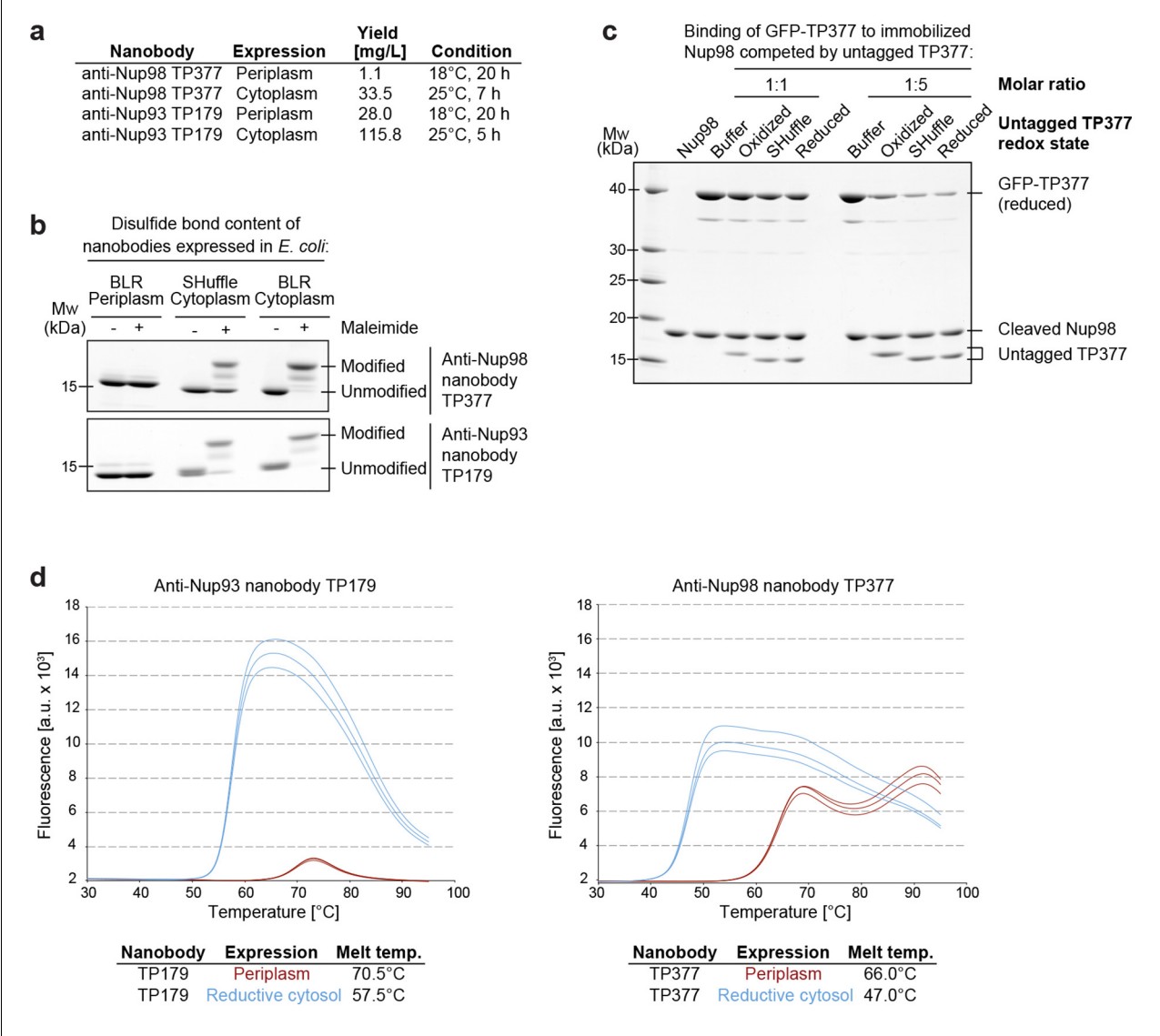

**Figure 1.** Affinity and thermostability of reduced and oxidized nanobodies. (**a**) Comparison of typical yields for the anti-Nup93 nanobody TP179 and the anti-Nup98 nanobody TP377 expressed either in the *Escherichia coli* BLR periplasm with a C-terminal His$_6$-tag or in the oxidative cytoplasm of *E. coli* SHuffle with an N-terminal His$_{14}$-bdNEDD8-tag. (**b**) Analysis of disulfide bond content using a maleimide shift assay. Anti-Nup93 nanobody TP179 and anti-Nup98 nanobody TP377, expressed either in the oxidative periplasm of *E. coli* BLR, the oxidative cytosol of *E. coli* SHuffle or in the reductive cytoplasm of *E. coli* BLR, were subjected to modification with biotin-PEG$_{23}$-maleimide in SDS–PAGE sample buffer (-DTT) and analyzed by non-reducing SDS–PAGE followed by Coomassie staining. (**c**) The redox state of the anti-Nup98 nanobody TP377 does not affect the affinity for its target. Biotinylated His$_{14}$-Avi-bdSUMO-tagged Nup98$^{716-866}$ was immobilized on Streptavidin agarose und used to bind the reduced GFP-tagged TP377. Binding was in the absence or presence of an equimolar amount or fivefold excess of nanobody competitor, namely untagged TP377 produced either in the oxidative periplasm, in the mildly oxidative cytoplasm of *E. coli* SHuffle or in the reductive cytoplasm of BLR. Bound nanobodies were then eluted by proteolytic cleavage of the bdSUMO tag of Nup98 and analyzed by SDS–PAGE followed by Coomassie staining. Note that the oxidized, disulfide bond-stabilized nanobody (produced in the periplasm) behaved like the reduced variant (produced in the *E. coli* BLR cytoplasm). Formation of the disulfide bond therefore does not seem to significantly contribute to the overall affinity. (**d**) Differential scanning fluorimetry (thermofluor, *Niesen et al., 2007*) analysis of nanobodies expressed in the oxidative periplasm (red) or the reductive cytosol (blue) of *E. coli* BLR. The anti-Nup93 and anti-Nup98 nanobodies were heated in the presence of Sypro Orange dye from 30 to 100°C and thermal unfolding curves were obtained. The melting temperature is derived from the inflection point of the curve.

In order to test whether disulfide bond formation can occur in the cytoplasm and if this is important for nanobody function, we expressed an anti-Nup93 and an anti-Nup98 nanobody either in the periplasm, the reductive cytoplasm of *E. coli* BLR or in the cytoplasm of *E. coli* SHuffle. The latter

strain contains a cytoplasmic disulfide isomerase and harbors mutations that render its cytoplasm (moderately) oxidative (*Lobstein et al., 2012*). The obtained nanobody variants were then treated with biotin-PEG$_{23}$-maleimide under denaturing conditions. Reduced nanobodies are thereby modified at their free cysteines, and the resulting size shift distinguishes them from disulfide-containing nanobodies (*Figure 1b*). While periplasmic secretion resulted in fully oxidized nanobodies, only a fraction of the SHuffle-expressed nanobodies contained a disulfide bond. Cytoplasmic expression in *E. coli* BLR yielded completely reduced nanobodies.

One could assume that the antigen affinity of nanobodies is negatively affected by a loss of their scaffold disulfide bond. A competition for antigen-binding revealed, however, no affinity difference between reduced and disulfide bond-containing anti-Nup98 nanobodies (*Figure 1c*). As expected, we observed by differential scanning fluorimetry (*Niesen et al., 2007*) a decreased thermostability of fully reduced anti-Nup98 and anti-Nup93 nanobodies (*Figure 1d*). Their melting temperatures of 47°C and 57°C are, however, still well above any reasonable incubation temperature for downstream applications.

All nanobodies that we obtained via phage display against a wide range of antigens could be functionally produced in the *E. coli* cytoplasm. Only very few of those nanobodies contained a second pair of cysteines that can form an additional, solvent-exposed disulfide bond between the antigen-binding loops CDR II and CDR III which likely contributes to the overall affinity (*Govaert et al., 2012*). However, most biochemical applications as well as imaging techniques like STORM require reducing conditions that disrupt accessible disulfide bonds, making such nanobodies a poor option anyway.

## Native purification of endogenous protein complexes using nanobodies

Cytoplasmic expression of nanobodies provides a number of advantages. First, the yield often exceeds 100 mg per liter of culture and can be up to 30 times higher as compared to periplasmic expression (*Figure 1a*). Second, it saves hands-on time, because a cumbersome preparation of a periplasmic fraction is bypassed, and third, a far broader range of fusion modules can be used.

We exploited this for affinity purification of endogenous target protein complexes with nanobodies and native elution. For this strategy, we produced His$_{14}$-Avi-(GlySer)$_9$-SUMOStar-(GlySer)$_9$-nanobody fusions and purified them by Ni$^{2+}$ chelate affinity chromatography and imidazole elution (*Figure 2—figure supplement 1a*). The Avi-tag can be biotinylated by cytoplasmic co-expression of the biotin ligase BirA in *E. coli* (*Schatz, 1993*; *Beckett et al., 1999*) or in vitro using the purified enzyme (*Fairhead and Howarth, 2015*). It then mediates binding of the purified nanobody to streptavidin magnetic beads. The interspersed long unfolded Gly-Ser spacers minimize steric hindrance effects. The SUMOStar module is an engineered SUMO variant that cannot be cleaved by endogenous eukaryotic desumoylases but by an engineered SUMOStar protease (LifeSensors), (*Liu et al., 2008*). In combination, these modules allow native elution of nanobody-bound target proteins or protein complexes by cleaving the tag with nanomolar concentrations of the SUMOStar protease. This strategy also provides a purer and more specific end product, because any protein species, which sticks non-specifically to the beads, will not be released. Thus, such highly specific protease elution makes the otherwise crucial control for matrix background-binding (*Marcon et al., 2015*; *Mellacheruvu et al., 2013*) essentially dispensable.

As a proof of principle, we purified five nucleoporin complexes from a *Xenopus* egg extract to near homogeneity (*Figure 2a and b*). For each complex we achieved a ≈10 000-fold enrichment within a single native purification step and yields of around 50%. The anti-Nup85 nanobody retrieved the ≈750 kDa nine-membered Y-complex as well as Tpr and Elys as specific but sub-stoichiometric binding partners. We obtained substantial amounts of the complex, namely 50–100 µg from as little as 2 ml egg extract, which initially contained ≈150 µg or ≈100 nM of the complex (*Wühr et al., 2014*). Post-elution with SDS sample buffer indicated a quantitative proteolytic release of the complex from the beads (*Figure 2—figure supplement 1b*).

The anti-Nup155 nanobody retrieved Nup155 as a single species. This might appear surprising as Nup155 is thought to contact Nup93 and Nup53/35 within the inner ring of the NPC scaffold (*Hawryluk-Gara et al., 2005*; *Hawryluk-Gara et al., 2008*; *Sachdev et al., 2012*). We therefore assume that mitotic post-translational modifications transiently suppress interactions between these proteins. We also purified the Nup98•Gle2 and the Nup62•Nup58•Nup54 complex using anti-Nup98 and anti-Nup54 nanobodies, respectively. Here, we included a RanQ69L•GTP wash to release

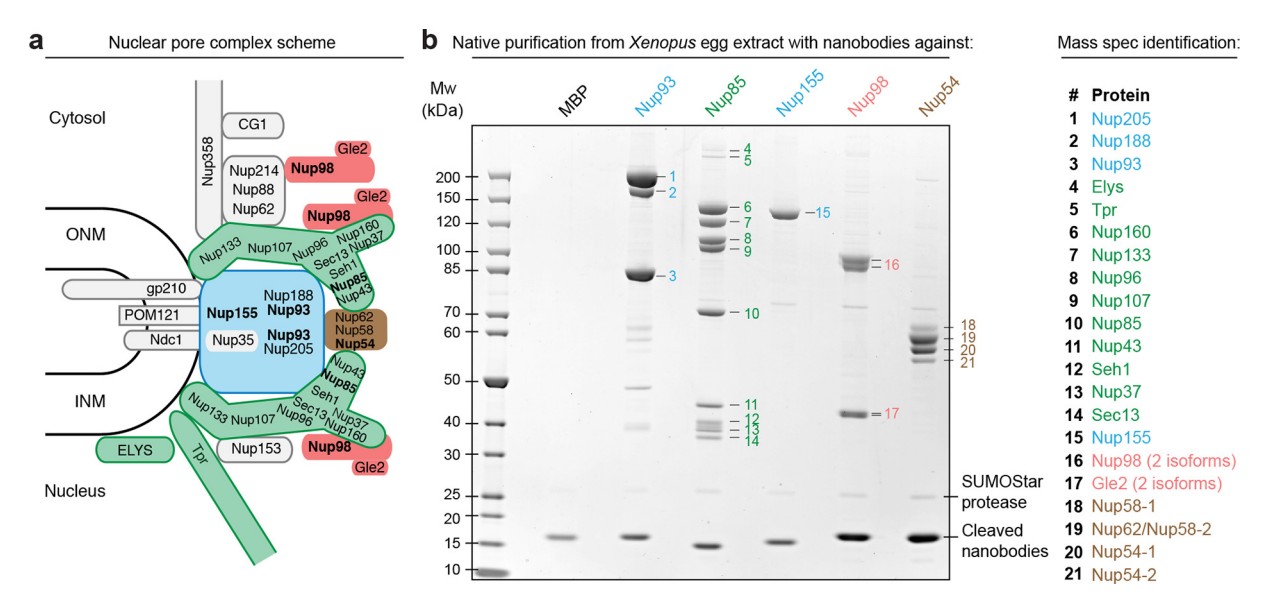

**Figure 2.** Purification and native elution of NPC subcomplexes with specific nanobodies. (a) Schematic representation of the subcomplex organization and relative localization of Nups within an asymmetric unit of the eightfold rotational symmetric vertebrate NPC (ONM/INM = outer and inner nuclear membrane). The nuclear and cytoplasmic rings of the structural NPC scaffold are mainly composed of the Nup107-Nup160 Y-shaped complex (green). The central inner ring of the scaffold is composed of the Nup93 subcomplex (blue). The scaffold is bound to the nuclear envelope via transmembrane Nups and further anchors FG-repeat nucleoporins (e.g. Nup98 [red] and the Nup62•Nup58•Nup54 complex [brown]) within the central channel, where they form the permeability barrier. Nups against which nanobodies were raised are highlighted in bold. (b) Native purification of major NPC scaffold subcomplexes and FG-repeat nucleoporins from *Xenopus* egg extract. Biotinylated His$_{14}$-Avi-(GlySer)$_9$-SUMOStar-(GlySer)$_9$-tagged nanobodies were immobilized on magnetic Streptavidin beads and then incubated with *Xenopus* egg extract. After washing, nanobodies were gently eluted along with their bound target complexes by SUMOStar protease cleavage. One tenth of the eluates were analyzed by SDS–PAGE and Coomassie staining. All labeled bands were identified via mass spectrometry. The color code represents the subcomplex organization of the NPC as illustrated in (a). A nanobody raised against *Escherichia coli* Maltose-binding protein (MBP) served as a negative control. NPC, nuclear pore complex.

The following figure supplement is available for figure 2:

**Figure supplement 1.** Optimization of native protein complex purification using nanobodies.

nuclear transport receptors, which otherwise would remain bound to the FG domains of Nup98 or the Nup62 complex (*Figure 2—figure supplement 1c*).

The anti-Nup93 nanobody purified the expected mixture of the two paralogous Nup93•Nup188 and Nup93•Nup205 complexes (*Theerthagiri et al., 2010*), which are also a part of the structurally least understood NPC inner ring. In this case, we analyzed the natively eluted material straightaway by negative stain electron microscopy (*Figure 3a*). Class averaging revealed characteristically curved α-solenoid fold-like particles, which are known to exhibit conformational flexibility (*Figure 3b*). The obtained structures were very reminiscent of the hook and eye-shaped structures reported earlier for the Nup188 and Nup205 orthologues from *Saccharomyces cerevisiae* (*Amlacher et al., 2011*) and *Myceliophthora thermophila* (*Andersen et al., 2013*). This suggests not only that the overall shape of the Nup93 complexes is conserved from fungi to vertebrates, but also that our single-step purification strategy for large protein complexes yields material of sufficient quality for a direct structural analysis.

## Site-specific fluorescent labeling of nanobodies

In order to use anti-Nup nanobodies to image their targets within intact NPCs, we initially modified them with NHS ester fluorophores. We found, however, that such NHS-labeled nanobodies performed remarkably poorly, in particular when far-red fluorophores were used. As documented by the specific examples below, none of the NHS-labeled nanobodies had sufficient probe quality to

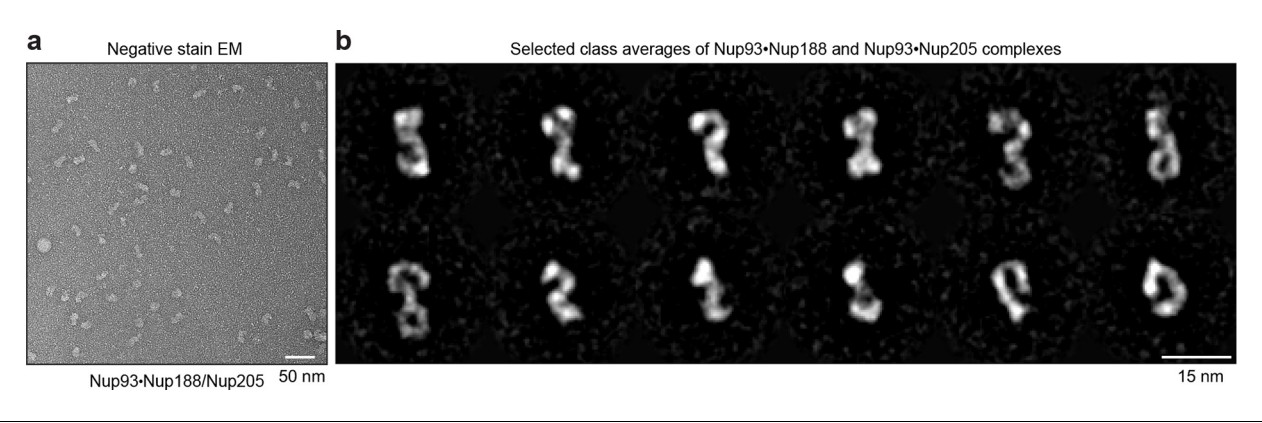

**Figure 3.** Structural analysis of natively purified Nup93 complexes. (a) Anti-Nup93 nanobody TP179-purified Nup93•Nup188 and Nup93•Nup205 complexes were subjected to the GraFix procedure (*Kastner et al., 2008*) and negative staining for analysis by electron microscopy. (b) Gallery of 12 selected class averages of Nup93•Nup188 and Nup93•Nup205 particles.

allow acquisition of STORM images. We therefore explored alternative and more reliable nanobody-labeling strategies.

One possibility was to label nanobodies at engineered (and reduced) cysteines with maleimides. This, however, posed the risk of modifying also the scaffold cysteines of the IgG-fold, which inevitably would cause an irreversible unfolding of the nanobodies.

To address this issue, we incubated reduced nanobodies with biotin-PEG$_{23}$-maleimide (*Figure 4*). After unfolding by urea, the scaffold cysteines became modified at either 37°C, 23°C, or 0°C. In native buffer, however, modification was quantitative only at 37°C, pointing to a transient exposure of the otherwise buried scaffold cysteines ('thermal breathing'). Importantly, they remained fully protected at 0°C, predicting that maleimide-labeling on ice would be fully selective for engineered surface cysteines.

In order to better guide cysteine placement in the nanobody framework, we solved the crystal structure of the anti-Nup98 nanobody TP377 in complex with the globular Nup98 NPC anchor domain (residues 716–866) at 1.9 Å resolution (*Figure 5a*, *Table 1*). TP377 contacts its target through all three CDR loops and does not block the absolute Nup98 C-terminus, which anchors Nup98 via Nup96 or Nup88 to the NPC scaffold (*Hodel et al., 2002*; *Griffis et al., 2003*; *Stuwe et al., 2012*). The internal disulfide bond-forming cysteines Cys22 and Cys96 of TP377 are reduced in the crystal structure.

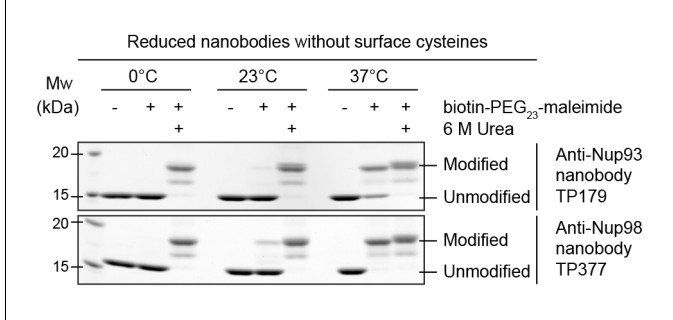

**Figure 4.** Maleimide modification of the internal cysteines of reduced nanobodies upon thermal unfolding. Indicated nanobodies, expressed in the reductive cytoplasm of *Escherichia coli* BLR, were incubated at the indicated temperatures in the presence or absence of a two-fold molar excess of biotin-PEG$_{23}$-maleimide (1.45 kDa) in buffer. The addition of 6 M urea served as a positive control for maleimide modification of the internal cysteines upon chemical unfolding.

We next mutated solvent-exposed small residues (Gly, Ser, and Ala) at six alternative positions of the nanobody scaffold to cysteines (*Figure 5b*; *Figure 5—figure supplement 1a*). We found that not only all individual mutants, but also nanobodies with up to three ectopic surface cysteines were well expressed and highly soluble in *E. coli* (*Figure 5—figure supplement 1b*). Moreover, cysteines on all six positions on our model anti-Nup98 nanobody TP377 could be quantitatively labeled with maleimide fluorescent dyes (*Figure 5c*). Even nanobodies carrying three fluorophores were readily obtained. Importantly, competitive binding assays indicated that the introduction of ectopic cysteines and their modification did not impair binding to the Nup98 target (*Figure 5—figure supplement 1c*). Based on the crystal structure, we estimate that fluorophores attached via an N-terminal cysteine or A75C to anti-Nup98 nanobody TP377 can be as close as ~2 nm to the target Nup98.

## Immunofluorescence with site-specifically labeled nanobodies

In order to test nanobodies in imaging, we grew *Xenopus laevis* XL177 cells on coverslips, digitonin-permeabilized their plasma membranes, incubated them with low nanomolar concentrations (1–10 nM) of labeled nanobody, and fixed them after several washing steps. In this workflow, even nanobodies with fixation-sensitive epitopes could bind their target.

We first tested anti-Nup98 nanobody TP377 carrying a single Alexa Fluor 647 maleimide at the six alternative positions (*Figure 5d*). In confocal laser scanning microscopy, all variants produced a very bright punctuate nuclear rim staining of XL177 cells, characteristic for NPCs, against a very low background.

Combining minimal label displacement with ease of cloning, we routinely labeled our nanobodies via an N-terminal cysteine. This way, all chosen NPC targets (Nup98, Nup93, Nup85 and Nup155) could be visualized with specific nanobodies carrying a single N-terminal Alexa Fluor 647 maleimide (*Figure 6a*). Despite the presence of only one dye molecule per nanobody, we again obtained very bright nuclear rim stains with very low background. Staining of Nup155 required a prior permeabilization with Triton X-100, probably because it is located in close proximity to the pore membrane and is likely buried by other NPC scaffold components (*Eisenhardt et al., 2014*; *Mitchell et al., 2010*).

For a direct comparison of the NHS chemistry for nanobody-labeling at lysines with maleimide-labeling at engineered surface cysteines, we chose Alexa Fluor 647 as a fluorophore and the widely used anti-GFP nanobody Enhancer (*Kirchhofer et al., 2010*) as an example (*Figure 6b*). When a HeLa Nup153-GFP cell line was stained, we observed a brilliant NPC signal for the Alexa Fluor 647 maleimide-labeled 'Enhancer', which perfectly coincided with the (weaker) GFP signal, and an extremely low background (*Figure 6c*). In contrast, when this nanobody was labeled at lysines with Alexa Fluor 647 NHS ester, it produced strong nucleoplasmic and cytoplasmic background staining, which essentially obscured the specific signal. The degree of labeling (DOL) was the same for both variants.

When the Alexa Fluor 647 NHS-labeled 'Enhancer' was applied to XL177 cells (which lack a GFP-target), we again observed very strong background (*Figure 6d*). In contrast, its Alexa Fluor 647 maleimide-labeled counterpart behaved like a perfect negative control. High background-staining was also observed with the Alexa Fluor 647 NHS-labeled anti-Nup98 nanobody TP377. The anti-Nup93 nanobody TP179 contains a lysine in CDR II and even lost antigen-binding after NHS modification. In contrast, the Alexa Fluor 647 maleimide-labeled anti-Nup98 and anti-Nup93 nanobodies behaved as perfect imaging reagents and gave crisp NPC signals against very low backgrounds. This comparison indicated that modification of (multiple) framework lysines likely creates hydrophobic patches that favor unspecific binding and aggregation. This is certainly sequence context-dependent and milder when reducing the labeling density. However, we did not observe any such complications when labeling nanobodies via engineered cysteines.

## Super-resolution imaging with site-specifically labeled nanobodies

Due to a diameter well below the diffraction limit, NPCs have been studied by super-resolution microscopy using either indirect immunofluorescence (*Löschberger et al., 2012*; *Göttfert et al., 2013*) or the anti-GFP nanobody (*Szymborska et al., 2013*). Site-specific fluorescent labeling of nanobodies via cysteines now reliably yields 'renewable' high-quality imaging reagents that can

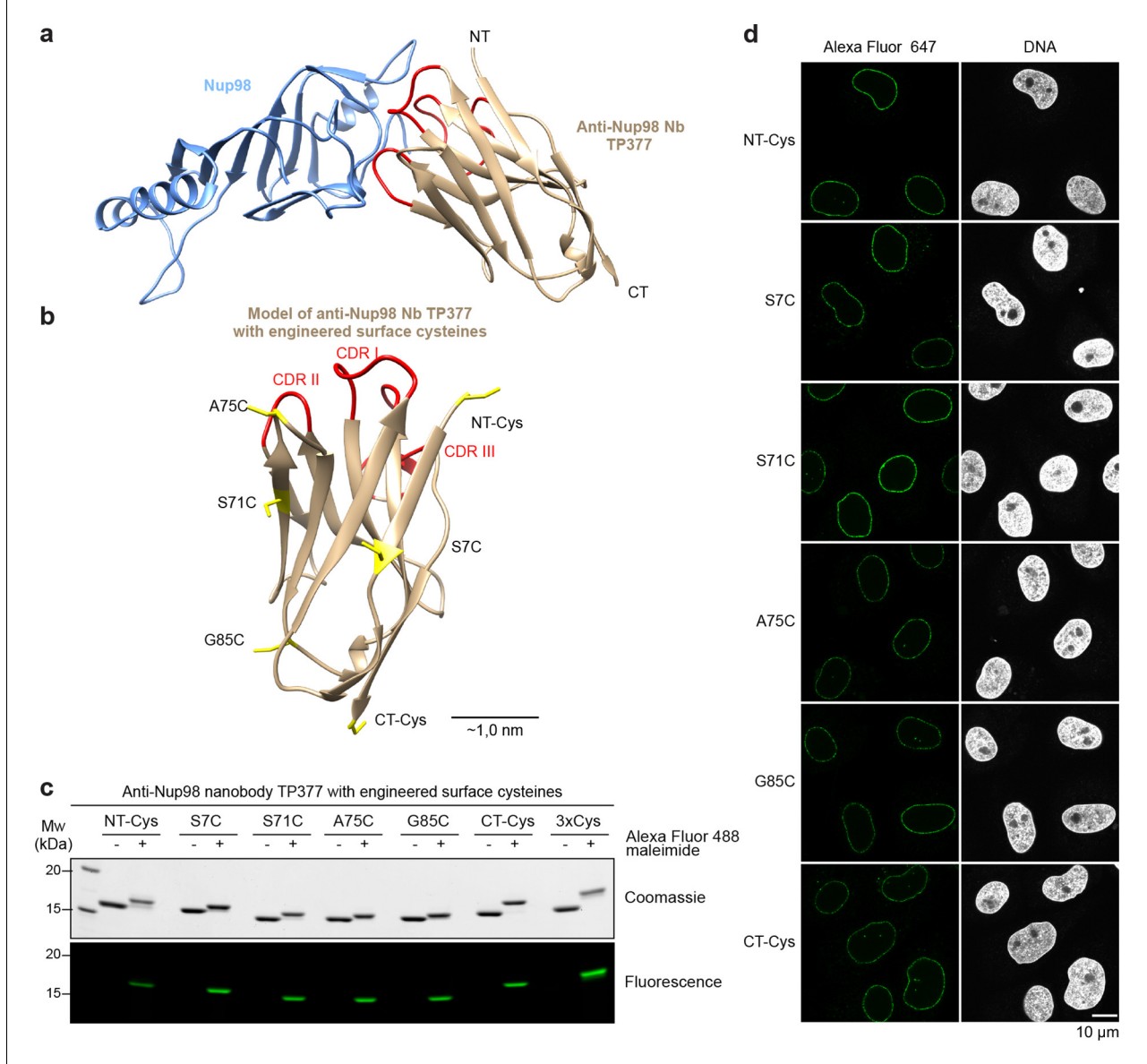

**Figure 5.** Site-specific fluorescent labeling of nanobodies. (**a**) Crystal structure of the Nup98 NPC anchor domain (Nup98$^{716-866}$, blue) in complex with the anti-Nup98 nanobody TP377 (beige). The three antigen-binding loops (CDR I-III) of TP377 are colored red. (NT = N-terminus, CT = C-terminus) (**b**) Tested positions of engineered cysteines (yellow) illustrated for nanobody TP377. Antigen-binding loops are shown in red. (**c**) Quantitative labeling of TP377 with Alexa Fluor 488 maleimide. TP377 with cysteines at the indicated positions can be quantitatively labeled with Alexa Fluor 488 maleimide. Labeling introduces a size shift in SDS–PAGE. Detection was either by Coomassie staining or by in-gel fluorescence. (3xCys = NT-Cys + S7C + S71C) (**d**) Digitonin-permeabilized *Xenopus* XL177 cells were incubated with 10 nM TP377 carrying a single Alexa Fluor 647 molecule at the indicated position. Cells were then washed, fixed, and counterstained with DAPI (DNA). A characteristic nuclear rim stain indicates labeling of NPCs. Note that labeling of TP377 very close to its antigen-binding loops did not perturb binding.

The following figure supplement is available for figure 5:

**Figure supplement 1.** Expression and relative affinity of anti-Nup98 nanobody TP377 with engineered surface cysteines.

bring fluorophores very close to their target. We therefore tested the performance of our anti-Nup nanobodies in STORM imaging (*Rust et al., 2006*) of XL177 cell NPCs (*Figure 7a–c*).

Interestingly, singly Alexa Fluor 647 maleimide-labeled anti-Nup nanobodies were sufficient to produce enough localizations to reconstruct very detailed views of individual NPCs, where multiple

**Table 1.** Crystallographic data collection and refinement statistics

| | Nup98•Nb TP377 complex[a] |
|---|---|
| **Data collection** | |
| Space group | $P4_1$ |
| Cell dimensions | |
| a, b, c (Å) | 66.59, 66.59, 87.90 |
| α, β, γ (°) | 90.00, 90.00, 90.00 |
| Resolution (Å) | 47.00-1.90 (1.95-1.90)[b] |
| $R_{sym}$ or $R_{merge}$ | 0.128 (>1)[b] |
| I / σI | 27.7 (2.6)[b] |
| Completeness (%) | 99.5 (98.7)[b] |
| Redundancy | 27.4 (27.4)[b] |
| | |
| **Refinement** | |
| Resolution (Å) | 47.00-1.90 |
| No. reflections | |
| Measured | 823105 |
| Unique | 30218 |
| $R_{work}$/$R_{free}$ | 0.167 / 0.196 |
| No. atoms | |
| Protein | 2176 |
| Water | 145 |
| Wilson B-factor (Å$^2$) | 27.4 |
| R.m.s. deviations | |
| Bond lengths (Å) | 0.010 |
| Bond angles (°) | 1.08 |
| Ramachandran statistics (%) | |
| Favored | 98.0 |
| Allowed | 2.0 |
| Outliers | - |

[a]A single crystal was used for data collection.
[b]Values in parentheses are for highest resolution shell.

copies of the imaged target proteins (Nup98, Nup93, and Nup155) appear arranged around the central NPC channel (*Figure 7c*). A whole nucleus stained with the model anti-Nup98 nanobody TP377 is shown in *Figure 7a* and magnified views of the nuclear envelope stained with anti-Nup93 and anti-Nup155 nanobody are shown in *Figure 7b*. Even after applying higher concentrations (~100–300 nM) of nanobody to saturate binding sites, we achieved very low background binding, indicating well-behaved imaging reagents.

## Rapid mapping of conformational epitopes via crosslinking mass spectrometry

Site-specifically labeled nanobodies enabled us to visualize their targeted epitope with high precision. Mapping the corresponding 'visible' epitopes would therefore reveal surface areas of the target that are accessible in a cellular environment. The complementary 'invisible' epitopes on the other hand, would hint to regions that are buried in interaction interfaces.

Epitope-mapping strategies based on binding assays to truncated or mutated antigens, co-crystallization or NMR observation of chemical shift perturbations are, however, not suited for high-throughput analysis or hardly applicable to conformational epitopes on protein complexes. We therefore considered crosslinking nanobodies to their target followed by sequencing of the crosslinked target peptide via mass spectrometry as a rapid epitope mapping strategy. Here, a crucial aspect is that a shorter crosslinker will provide a better spatial resolution, provided crosslinkable groups are in reach. As cysteines are by far the best crosslinkable groups, and because we had already placed cysteines at the nanobody surface in close proximity to bound targets, we assessed their suitability for epitope-mapping.

As a proof of principle, we crosslinked two anti-Nup93 nanobodies, with or without an N-terminal cysteine, to Nup93 using either an $NH_2$-to-$NH_2$ (Bis-NHS; BS3; 11.4Å) or an SH-to-$NH_2$ (Mal-NHS; BMPS; 5.9 Å) crosslinker (*Figure 8a*). For both anti-Nup93 nanobodies (TP179 and TP324), exclusive amine-crosslinking was very inefficient and produced only few nanobody•Nup93 adducts that run at higher molecular weight in SDS–PAGE. However, combining the N-terminal cysteine on the nanobody with the (far shorter) heterobifunctional crosslinker, produced very prominent nanobody•Nup93 crosslinks. Their position was then clearly identifiable by LC-MS/MS (*Figure 8b–c*, *Figure 8—figure supplement 1*).

For a better visualization of the positions of the identified nanobody crosslinks we generated a structural model of Nup93$^{168-end}$ using I-TASSER (*Zhang, 2008*), based on structures of its yeast ortholog (*Jeudy and Schwartz, 2007*; *Schrader et al., 2008*) (*Figure 8c*).

We used the anti-Nup93 nanobody TP179 in STORM imaging of Nup93 within the NPC and could now map its accessible epitope. TP179 binds to the middle region of the J-shaped structure of Nup93 surrounding residues K607 and K612 (*Figure 8—figure supplement 1a–b*), while TP324 has

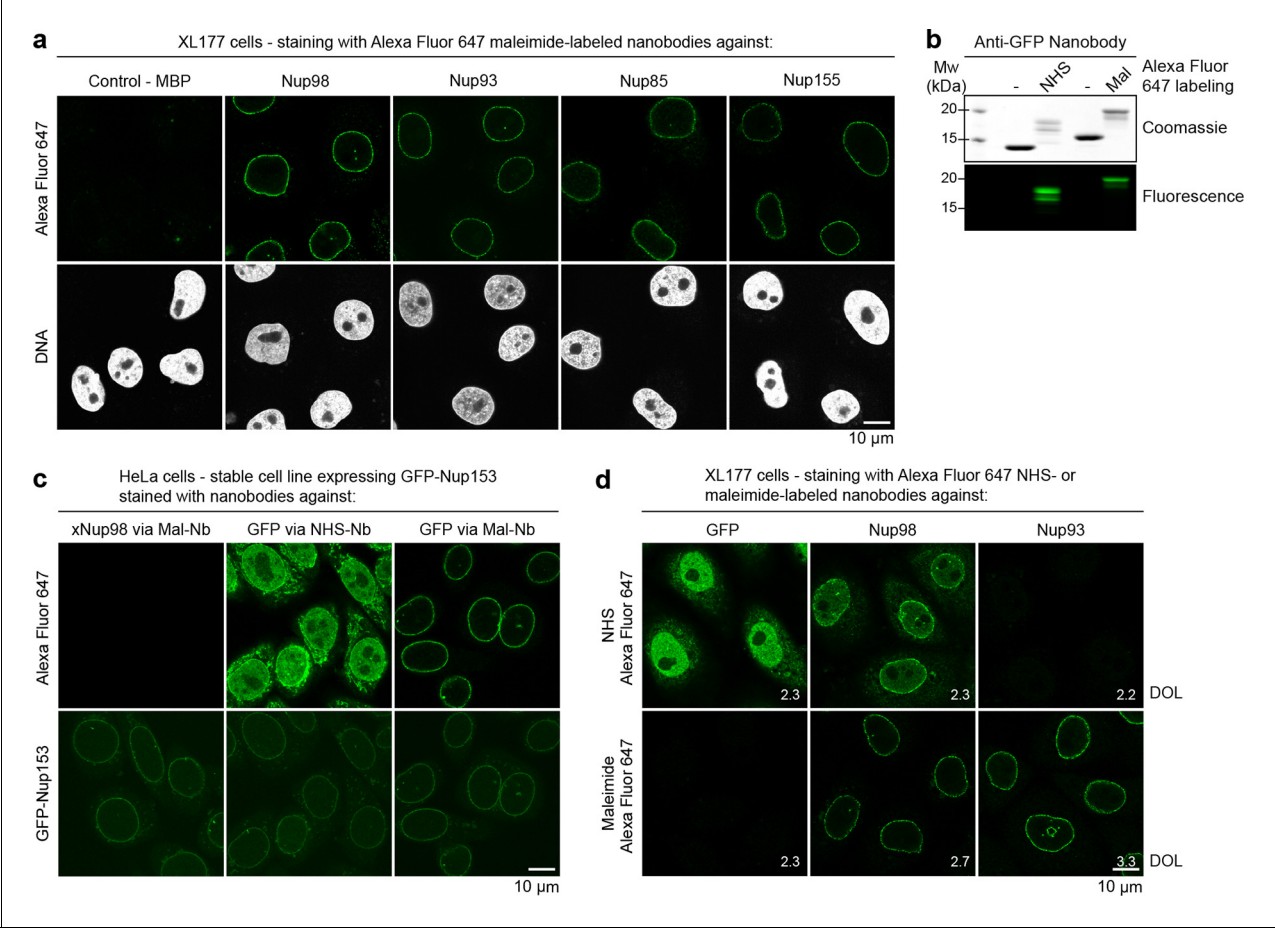

**Figure 6.** Immunofluorescence with site-specifically labeled anti-Nup nanobodies. (**a**) *Xenopus* XL177 cells were digitonin-permeabilized and stained with anti-Nup nanobodies carrying a single N-terminal Alexa Fluor 647 maleimide dye before fixation and DAPI staining. A characteristic nuclear rim stain indicates labeling of NPCs. A nanobody raised against *Escherichia coli* Maltose-binding protein (MBP) served as a negative control. (**b**) Labeling of the anti-GFP nanobody Enhancer with Alexa Fluor 647 NHS ester at lysines or at three engineered cysteines using Alexa Fluor 647 maleimide. Labeling introduces a size shift in SDS–PAGE. Detection was either by Coomassie staining or by in-gel fluorescence. (**c**) Staining of HeLa cells stably expressing GFP-tagged Nup153 with the anti-GFP nanobody labeled via NHS ester or maleimide Alexa Fluor 647. The nanobody TP377, raised against *Xenopus* (*x*) Nup98, does not cross-react with human Nup98 and served as a negative control. The NHS-labeled GFP nanobody produced strong background-staining, while its maleimide-labeled version yielded bright nuclear rim stains. (**d**) Staining of XL177 cells with nanobodies labeled with Alexa Fluor 647 either at their internal lysine residues (NHS ester dye) or via engineered cysteines (maleimide dye). Note that the widely used anti-GFP nanobody Enhancer produces significant background staining when labeled via lysines but not when using engineered cysteines and a maleimide dye. All nanobodies were used at a concentration of 10 nM and all images were obtained under identical settings. DOL, degree of labeling.

a C-terminal epitope surrounding lysines K762, K765, and K782 of Nup93 (*Figure 8—figure supplement 1c–e*). The C-terminal region of Nup93 was previously shown to be essential for NPC assembly (*Sachdev et al., 2012*). Accordingly, anti-Nup93 nanobody TP324 that targets the C-terminus of Nup93 does not stain intact NPCs (data not shown), but rather represents a good candidate to selectively disrupt NPC assembly.

## Discussion

We developed a well-characterized toolset of high-affinity nanobodies against the vertebrate NPC and established novel strategies to use these nanobodies to natively purify large NPC subcomplexes and to reliably label them with fluorophores for precise super-resolution localization. While these nanobodies will be very valuable to the nucleocytoplasmic transport field, we expect the presented strategies to be widely applicable to all nanobodies.

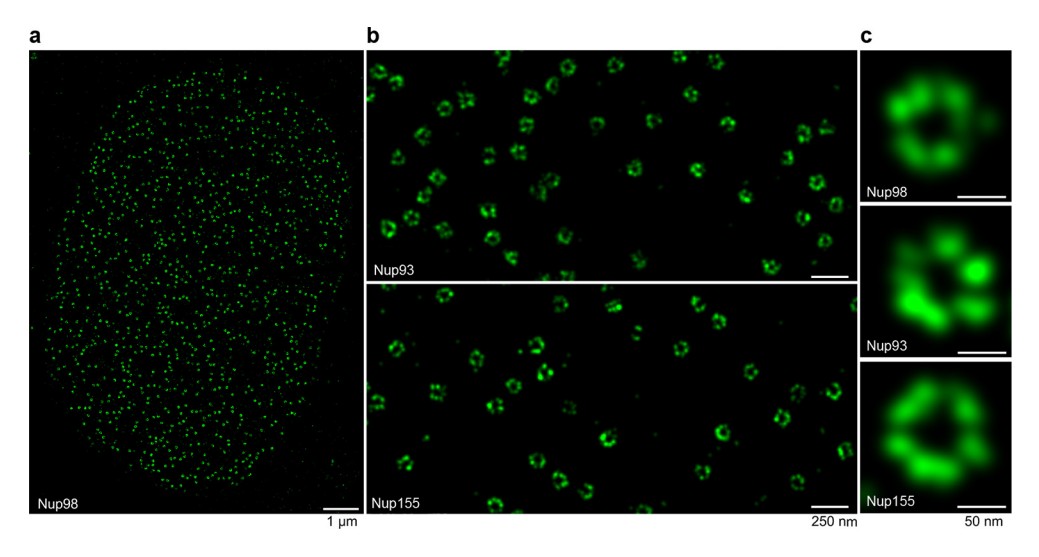

**Figure 7.** STORM imaging of nuclear pore complexes stained with site-specifically labeled anti-Nup nanobodies. (**a**) STORM image of an entire XL177 cell nucleus stained with anti-Nup98 nanobody TP377 carrying a single N-terminal Alexa Fluor 647 maleimide. (**b**) Close-up view of XL177 cell nuclear envelope regions stained with anti-Nup93 nanobody TP179 (upper panel) or an anti-Nup155 nanobody (lower panel) containing multiple NPCs. (**c**) STORM images of individual NPCs stained with indicated anti-Nup nanobodies.

Nanobodies against a single epitope of a larger protein complex now allow a native single-step purification of the entire complex, and thus a subsequent structural and functional analysis. This is certainly especially useful for complexes that are not directly accessible to recombinant production. Furthermore, nanobody-purified endogenous complexes can be used as antigens for another round of immunization, and binders to all complex components can then be selected from the successive nanobody library. Mapping epitopes via crosslinking mass spectrometry will become especially important when selecting nanobodies against such complex antigens (like large subcomplexes, whole organelles or vesicles) that cannot be made recombinantly. Combining these strategies therefore has the potential to significantly increase throughput in the selection and identification of renewable binders to eukaryotic proteomes (*Colwill et al., 2011*).

Finally, we introduced a method for a reliable fluorescent labeling of nanobodies using surface cysteines and maleimide chemistry. This way, we obtained well-behaved imaging reagents that can bring fluorescent dyes as close as 1–2 nm to their target. Maleimide-labeled nanobodies consistently recognized their antigens far better and produced less background than the corresponding NHS-modified variants. NHS esters have the additional disadvantage that they react not only with amino groups, but also rapidly hydrolyze in aqueous buffers. This makes it difficult to adjust labeling densities and requires adding them in substantial molar excess. In contrast, maleimide-labeling of exposed cysteines is quantitative even with just stoichiometric amounts of labeling reagent and thus far more economical.

Site-specific and quantitative fluorescent labeling of nanobodies is going to be crucial for super-resolution microscopy aiming at a detailed structural analysis or determination of absolute protein copy numbers. It also allows predicting the effective label displacement, a fact that will be especially important when applying particle averaging techniques to localization microscopy data (precision of <1 nm reported by *Szymborska et al., 2013*). Because of its well-defined dimension and symmetric structure, the NPC has become a benchmark for many new advancements of super-resolution microscopy (*Schermelleh et al., 2008*; *Szymborska et al., 2013*; *Göttfert et al., 2013*). The anti-NPC nanobodies described here excelled in super-resolution imaging; they can be renewably produced in high yields and are therefore ideal labeling reagents for such benchmark studies.

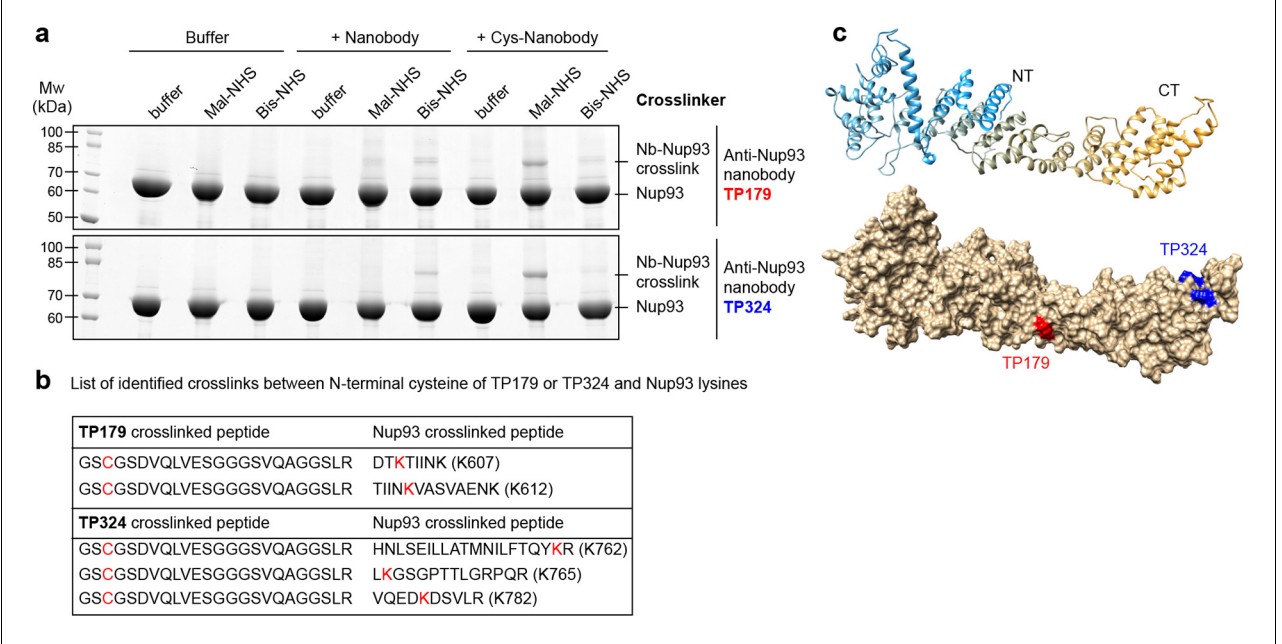

**Figure 8.** Rapid epitope mapping via crosslinking mass spectrometry. (a) Crosslinking of two different anti-Nup93 nanobodies (TP179 and TP324) to Nup93 using amine-to-amine ('Bis-NHS'; BS3; 11.4 Å linker length) or thiol-to-amine ('Mal-NHS'; BMPS; 5.9 Å linker length) crosslinking reagents. The combination of the very short Mal-NHS crosslinker with an engineered cysteine close to the antigen-binding loops provided for both nanobodies by far the highest yield of crosslinked nanobody•Nup93 adduct. (b) List of identified crosslinked peptides involving Nup93 lysines and Cys-TP179 or Cys-TP324. The crosslinked amino acids are highlighted in red (see also *Figure 8—figure supplement 1*). (c) Crosslinked lysines of Nup93 to the N-terminal cysteine on anti-Nup93 nanobodies TP179 (red) or TP324 (blue) are depicted on a structural model of Nup93$^{168\text{-end}}$ generated by I-TASSER (*Zhang, 2008*). Based on the orthologous yeast crystal structures (*Jeudy and Schwartz, 2007*; *Schrader et al., 2008*), Nup93 is predicted to form a similar J-shaped structure (color gradient: NT = N-terminus in blue to CT = C-terminus in orange). Whereas TP179 binds to the central portion, TP324 binds to the C-terminus of Nup93.

The following figure supplement is available for figure 8:

**Figure supplement 1.** Representative MS/MS spectra of the crosslinked peptides derived from Nup93•nanobody complexes.

## Methods

### Alpaca immunization

Two female alpacas, held at the Max Planck Institute for Biophysical Chemistry, were immunized with 0.5–1.0 mg protein or protein complex at 3–4 week intervals for 3–4 times. The antigens had been expressed recombinantly in *E. coli*, affinity-purified and mixed with a mild squalen/α-tocopherol/Tween-80-based adjuvant (oil-in-water emulsion) before immunization. In detail, we immunized one animal with xlNup93$^{168\text{-end}}$, xtNup98$^{716\text{-866}}$, and the xlNup62$^{342\text{-547}}$•Nup58$^{267\text{-490}}$•Nup54$^{146\text{-535}}$ complex (*Chug et al., 2015*) and another animal with full length xlNup155 and full length xlNup85.

### Nanobody library generation and phage display selection

Four days after the final boost, 100 ml of blood were collected from the immunized animal. Peripheral blood lymphocytes were isolated by density gradient centrifugation using Leucosep tubes (Greiner Bio-One, Austria) and total RNA was prepared according to Chomczynski and Sacchi (*Chomczynski and Sacchi, 2006*). For library generation, cDNA was generated from 30 μg of total RNA using the Superscript III kit (Life Technologies) with an IgG-CH2 domain specific primer, pCALL002 (*Conrath et al., 2001*). For VHH domain amplification, a nested PCR was performed. The first PCR product was obtained using the primers AlpVh-L, AlpVHHR1 and AlpVHHR2 (*Maass et al., 2007*), which anneal in the leader sequence and the VHH-specific hinge regions. The first PCR product served as a template for amplification with VHH framework 1 and framework 4-specific primers

(PT411: AATATA<u>GGATCC</u>CAAGTGCAGCTCGTRGAGTCTGG and 38: GGACTAGT<u>GCGGCCGC</u>TG-GAGACGGTGACCTGGGT) introducing BamHI and NotI restriction sites (underlined), respectively. A previous study (*Rothbauer et al., 2006*) used NcoI, which according to our sequencing data very frequently cleaves within the CDR I-coding region, resulting in many truncated non-functional nanobody sequences. The BamHI and NotI digested VHH immune library was then cloned into a pHEN4-derived phagemid (*Arbabi Ghahroudi et al., 1997*) and used to transform *E. coli* TG1 (Lucigen). A library of 2–3 x $10^8$ individual transformants was infected with helper phage M13KO7 (New England Biolabs) and VHH-displaying bacteriophages were produced overnight while shaking at 37°C. Bacteriophages were purified from the culture supernatant by two successive precipitation steps with 4% PEG-8000, 500 mM NaCl. The pellets were gently resuspended in 50 mM Tris/HCl pH 7.5, 300 mM NaCl and the obtained phage stock solution used directly for selection. Panning was performed using recombinant antigens carrying an Avi-tag that was biotinylated in *E. coli* by co-expression of biotin ligase BirA (*Beckett et al., 1999*; *Schatz, 1993*). For the first round of panning, biotinylated antigen was pre-immobilized on Dynabeads Streptavidin T1 (Life Technologies). During later rounds, phages were incubated with biotinylated antigen in solution and then retrieved by adding magnetic beads. After thorough washing, bound phages were eluted and the obtained binders were characterized. Typically three rounds of panning with decreasing antigen concentration (e.g. 100 nM, 20 nM, and 1 nM) and increasingly thorough washing were performed.

## Expression and purification of nanobodies

Nanobodies with protease-cleavable affinity tags or engineered cysteines were routinely expressed in the cytoplasm of *E. coli* BLR (BL21 derivative; Novagen) or *E. coli* SHuffle Express (New England Biolabs). A 50 ml preculture (Terrific Broth or 2YT medium containing 50 µg/ml Kanamycin) was grown overnight at 28°C. The culture was then diluted with fresh medium to 250 ml. After 1 h of growth at 25°C, protein expression was induced for 3–5 h by adding 0.2 mM IPTG. After addition of 1 mM PMSF and 10 mM EDTA to the culture, bacteria were harvested by centrifugation, resuspended in lysis buffer (50 mM Tris/HCl pH 7.5, 300 mM NaCl, 20 mM imidazole) and then lysed by sonication. The lysate was cleared by ultracentrifugation for 1.5 h (T647.5 rotor, Sorvall, 38,000 rpm) at 4°C.

For native affinity purification, nanobodies were fused to an N-terminal $His_{14}$-Avi peptide (GLNDI-FEAQKIEWHE)-(GlySer)$_9$-scSUMOStar-(GlySer)$_9$-tag and co-expressed with the biotin ligase BirA (*Beckett et al., 1999*; *Schatz, 1993*) in the presence of ~20 µg/ml biotin in the medium. Following lysis, nanobodies were purified by $Ni^{2+}$ chelate affinity chromatography. After washing with lysis buffer, the bound protein was eluted with 50 mM Tris/HCl pH 7.5, 300 mM NaCl, 500 mM imidazole. Alternatively, the purified enzyme BirA was added after binding the nanobody to a $Ni^{2+}$ chelate affinity resin for on-column biotinylation in Bio-buffer (50 mM Tris/HCl pH 7.5, 100 mM NaCl, 10 mM ATP, 12.5 mM $MgCl_2$, 250 µM biotin). For this, 1 µM BirA in twofold resin bed volume of Bio-buffer was incubated with resin under constant mixing for 2 h at room temperature. Nanobodies with engineered cysteines carried an N-terminal $His_{14}$-bdNEDD8-tag and were affinity purified via $Ni^{2+}$ chelate affinity chromatography. After washing, untagged nanobodies were eluted by cleavage with the bdNEDP1 protease (*Frey and Görlich, 2014*).

## Native purification of protein complexes from *Xenopus* egg extract

Interphase low-speed supernatant (LSS) extract was prepared from *Xenopus* eggs essentially as described before (*Blow and Laskey, 1986*) and stored at -80°C. LSS was thawed, diluted fourfold in S250 buffer (20 mM HEPES pH 7.5, 90 mM KAc, 2 mM MgAc, 250 mM sucrose), supplemented with 5 mM ATP and 5 µg/ml Cytochalasin B and then centrifuged in Seton tubes (SETON Scientific) for 1 h at 235,000 g in a Sorvall Discovery M120 SE ultracentrifuge (S52ST rotor). The lipid- and membrane-free high-speed supernatant (HSS) extract was retrieved by puncturing the side of the tube with a needle and served as starting material for affinity purifications.

Biotinylated nanobodies were immobilized on magnetic Dynabeads MyOne Streptavidin T1 (Life Technologies) in S250 buffer for 30 min at 4°C. Remaining biotin-binding sites on Streptavidin were subsequently blocked by incubation with 50 µM Biotin-PEG-COOH (Iris Biotech) in S250 buffer for 15 min. The blocked beads were then added to *Xenopus* egg extract (= HSS) for 1 h at 4°C. Using a magnetic rack, the beads were separated from extract and washed twice in S250 buffer, followed by

two washes in 50 mM Tris/HCl pH 7.5, 300 mM NaCl, 0.05% Tween-20. Nanobody•target protein complexes were then eluted by adding 0.5 µM SUMOStar protease (*Liu et al., 2008*) in 50 mM Tris/HCl pH 7.5, 300 mM NaCl for 45 min at 4°C.

## Electron microscopy sample preparation and image processing

Directly after elution, nanobody-purified Nup93•Nup188 and Nup93•Nup205 complexes were subjected to the GraFix protocol (*Kastner et al., 2008*) for complex stabilization. Briefly, ~200 pmoles of nanobody-purified complexes (~140 µl) were loaded onto a 4.2 ml 5% – 20% (w/v) sucrose-gradient supplemented with 0.1% (v/v) glutaraldehyde in the 20% fraction. The gradient was run in a TH-660 ultra-centrifuge rotor (Thermo Scientific; 34,000 rpm, 16 h, 4°C) and then fractionated into 200 µl fractions. The chemically stabilized molecules from the peak fraction were adsorbed to a thin carbon film by surface flotation for 1 min and negatively stained in uranyl formate solution. Images were acquired at room temperature at a magnification of 117,333× on a 4k x 4k CCD camera (TVIPS GmbH) using twofold pixel binning (2.5 Å/pixel) in a Philips CM200 FEG electron microscope (Philips/FEI) operated at 160 kV acceleration voltage. From the images, 8139 particles were selected (*Busche, 2013*) and subjected to contrast transfer function correction (*Sander et al., 2003*). Subsequently, an initial alignment-by-classification (*Dube et al., 1993*) step followed by iterative multi-reference alignment and multivariate statistical analysis were performed using IMAGIC (*van Heel et al., 1996*), resulting in 2D class averages.

## Site-specific fluorescent labeling of nanobodies with engineered cysteines

Purified nanobodies with engineered cysteines were freshly reduced by adding 15 mM TCEP for 10 min on ice. Using PD-10 desalting columns (GE Healthcare), the buffer was exchanged to Maleimide-labeling buffer (100 mM potassium phosphate pH 6.4, 150 mM NaCl, 1 mM EDTA, 250 mM sucrose) that had been vacuum degased and purged with argon. For a standard labeling reaction, 10 nmoles of nanobody (concentration 75–150 µM) were rapidly mixed with 12 nmoles of Alexa Fluor 647 C2 Maleimide (Life Technologies) (from a 20 mM stock in DMF), neutralized to pH 7.5 with $K_2HPO_4$ and incubated for 1.5 h on ice. Free dye was separated from labeled nanobody by buffer exchange to Maleimide labeling buffer on PD10 desalting columns. Quantitative labeling was quality controlled by calculating the degree of labeling (DOL), which defines the molar ratio of dye to protein, as well as by SDS–PAGE and Coomassie staining. In order to obtain nanobodies with three fluorophores, we recommend introducing cysteines at the N-terminus, Ser7 and Ala75 (other amino acids can occur at these positions in different nanobodies) of a given nanobody sequence to achieve the smallest possible label displacement. For easy cloning, three cysteines can also be introduced with primers in a single PCR reaction (positions: N-terminus and Ser7 in the forward primer and at the C-terminus with the reverse primer).

For Alexa Fluor 647 NHS-labeling, 10 nmoles of nanobody (concentration 75–150 µM) were incubated with an eightfold molar excess of dye (20 mM stock in DMF) in 100 mM sodium bicarbonate pH 7.8, 300 mM NaCl for 1 h at 23°C. Subsequently, the reaction was quenched and free dye was separated by buffer exchange to 50 mM Tris/HCl pH 7.5, 300 mM NaCl, 250 mM sucrose on PD10 desalting columns.

## Immunofluorescence with fluorescent nanobodies

*Xenopus laevis* XL177 epithelial cells (*Miller and Daniel, 1977*; *Ellison et al., 1985*) were grown on coverslips at 27°C with 5% $CO_2$ in *Xenopus* culture medium: (25% v/v water, 10% fetal bovine serum, 65% DMEM high glucose medium containing pyruvate and glutamine, and 50 U/ml penicillin + 50 µg/ml streptomycin). Alternatively *Xenopus laevis* A6 cells (#ATCC CCL-102 [TM]) can be used. Cells were pre-fixed for 30 s with 2.4% (w/v) paraformaldehyde in Transport buffer (TRB) (20 mM HEPES pH 7.5, 5 mM MgAc, 110 mM KAc, 1 mM EGTA, 250 mM sucrose) to prevent detachment of cells from the coverslips and briefly washed twice with TRB. The cells were then permeabilized for 8 min on ice with pre-chilled TRB containing 25 µg/ml Digitonin. Following two washes with TRB + 1% (w/v) Bovine Serum Albumin (BSA) for 5 min each, the cells were incubated with 1–10 nM of fluorescent nanobody for 15 min on ice. Subsequently, the cells were washed thrice for 5 min with TRB + 1% (w/v) BSA at room temperature and then fixed for 10 min with 3% (w/v) paraformaldehyde in TRB. The

nuclear envelopes of the fixed cells were afterwards permeabilized with 0.3% Triton X-100 for 3 min, washed with 1xPBS and DNA was stained by addition of 2 µg/ml DAPI in 1xPBS for 10 min. The coverslips were mounted in SlowFade Gold or SlowFade Diamond Antifade Mountant (Life Technologies) and analyzed by confocal laser-scanning microscopy on a Leica SP5 microscope.

## STORM imaging of nanobody-stained XL177 cells

In order to obtain the highest labeling efficiency, XL177 cells were stained with Alexa Fluor 647 maleimide-conjugated nanobodies initially after a short pre-fixation and digitonin permeabilization of the plasma membrane. The cells were subsequently fixed, the nuclear envelope was permeabilized with Triton X-100 and labeled nanobodies were added again. The optimal concentration of each nanobody for both steps was titrated before, using confocal microscopy. All STORM imaging experiments were carried out in MEA imaging buffer as previously described (*Dempsey et al., 2011*). The buffer consisted of 50 mM Tris/HCl pH 8.0, 10 mM NaCl, 10% glucose (w/v), 10 mM β-mercaptoethylamine pH 8.5 (Sigma, 30070), and 1% of an enzymatic oxygen scavenger system stock solution, added to the buffer immediately prior to use. The oxygen scavenger stock solution was prepared by mixing glucose oxidase (10 mg, Sigma, G2133) with catalase (50 µl, 20 mg/ml$^{-1}$, Sigma, C30) in 1x PBS (200 µl), and centrifuging the mixture at 13,000 rpm for 1 min.

STORM imaging measurements were performed using a custom-built STORM microscope, based on an inverted fluorescence microscope stand (Olympus IX71) as previously described (*Dempsey et al., 2011*). The microscope was fitted with a 100x oil-immersion objective lens (Olympus UPLANSAPO, NA1.4), which enabled efficient detection of single fluorophores. The objective lens was mounted on a piezo-positioner (Piezo Jena), which enabled fine focus adjustment. A custom-built focus-lock system was used to maintain a stable focus during data acquisition. For STORM imaging, photo-switchable Alexa Fluor 647 was excited at 642 nm, and in some measurements, the sample was also exposed to 405 nm light to increase the activation rate of switching. A fiber laser (MPB Communications, 2RU-VFL-P-1000-642) was used to generate 642 nm light. The laser illumination was configured such that the illumination angle could be varied between an epi-illumination geometry and a total internal reflection (TIRF) illumination mode. Typically, the sample was illuminated with oblique illumination (not TIRF) for reduced background signal. Fluorescence emission of Alexa Fluor 647 was detected using an EMCCD camera (Andor Ixon DU860). STORM data analysis was carried out using custom analysis software, as previously described (*Bates et al., 2007*).

## Crystallization and structure determination

The *Xenopus* Nup98$^{716-866}$ NPC anchor domain and the anti-Nup98 nanobody TP377 were expressed with an N-terminal His$_{14}$-bdSUMO-tag and purified using Ni$^{2+}$ chelate affinity chromatography. Crystallization required an exchange of the surface-exposed cysteine 821 of Nup98 to serine. Highly pure untagged protein was cleaved off the column using 50 nM bdSEN1P protease (*Frey and Görlich, 2014*) in 20 mM Tris, 20 mM NaCl. The complex was formed by incubating equimolar amounts of Nup98$^{716-866}$ and TP377 o/n at 4°C and then subjected to anion exchange chromatography using a HiTrap Q HP 5 ml column (GE Healthcare). The eluted complex was then further purified using gel filtration on a Hi-Load Superdex 75 16/60 column equilibrated in 20 mM Tris/HCl pH 7.5, 50 mM NaCl. The complex was crystallized by the vapor diffusion method in sitting drops. 60 nl of a reservoir solution containing 45% (w/v) Pentaerythritol propoxylate (17/8 PO/OH; Jena Bioscience) and 100 mM Tris pH 8.5 was mixed with 60 nl of the prepared protein complex solution concentrated to 25 mg/ml. Crystals grew within 1 day at 20°C and were flash-frozen in liquid nitrogen without additional cryo-protection. Diffraction data were collected at 100 K with a wavelength of 0.9787 Å on the beamline PXII at the Swiss Light Source (SLS) at the Paul Scherrer Institute, Switzerland. Crystals belonged to the space group P4$_1$ and diffracted to 1.9 Å (see *Table 1*). For structure determination, molecular replacement was performed in PHASER with a published nanobody structure (PDB 4KRN; *Schmitz et al., 2013*) as a search model. The resulting electron density map was used for automated model building in Phenix (*Adams et al., 2010*).

## Epitope mapping via crosslinking mass spectrometry

Anti-Nup93 nanobodies TP179 or TP324 and Nup93 (~20 µM each) were incubated on ice for 30 min in Maleimide labeling buffer to allow complex formation. After adding 40 µM of crosslinking

agent, the pH was increased to 7.5 and the reaction was continued for 1 h on ice. The following crosslinkers were used 'Mal-NHS' = BMPS (3-[Maleimido]propionic acid NHS ester, Iris Biotech, CAS #55750-62-4) and 'Bis-NHS' = BS3 (Suberic acid bis[sulfo NHS ester], Life Technologies, CAS #82436-77-9). One-eighth of the reaction was loaded on a SDS–PAGE gel. The band corresponding to crosslinked products was excised and subjected to in-gel trypsin digestion as described (*Schmidt and Urlaub, 2009*). The peptide fragments were extracted in a solvent system containing 5% acetonitrile (ACN), 0.1% formic acid (FA) to a final volume of 20–30 µl and submitted to liquid chromatography-tandem mass spectrometry (LC-MS/MS) analysis.

For LC-MS/MS analysis, 6 µl of the sample solution was injected into a nano-liquid chromatography system (UltiMate$^{TM}$ 3000 RSLCnano system) including a 3 cm × 150 µm inner diameter C18 trapping column in-line with a 30 cm × 75 µm inner diameter C18 analytical column (both in-house packed with 1.9 µm C18 material, Dr. Maisch GmbH). Peptides were desalted on the trapping column for 3 min at a flow rate of 10 µl/min in 95% of mobile phase A (0.1% FA in $H_2O$, v/v) and 5% of mobile phase B (80% ACN and 0.05% FA in $H_2O$, v/v), eluted from the trapping column, and separated on the analytical column using a 43 min linear gradient of 15–46% mobile phase B at a flow rate of 300 nl/min. Separated peptides were analyzed on-line with an Orbitrap Fusion mass spectrometer (Thermo Scientific). The 20 most intense precursor ions with charge states 3–8 in the survey scan (380–1580 m/z scan range) were isolated in the quadrupole mass filter (isolation window 1.6 m/z) and fragmented in the higher energy collisional dissociation (HCD) cell with 30% normalized collision energy. A dynamic exclusion of 20 s was used. Both the survey scan (MS1) and the product ion scan (MS2) were performed in the Orbitrap at 120,000 and 30,000 resolution, respectively. Spray voltage was set at 2.3 kV and 60% of S-lens RF level was used. Automatic gain control (AGC) targets were set at $5 \times 10^5$ and $5 \times 10^4$ for MS1 and MS2, respectively.

## Database search for crosslinked peptides

The raw data of LC-MS/MS analysis were converted to mascot generic format (mgf) files by Proteome Discoverer 2.0.0.802 software (Thermo Scientific). The mgf files were searched against a FASTA database containing the sequences of the nanobody and Nup93 by pLink 1.22 software (*Yang et al., 2012*) using a target-decoy strategy. Database search parameters included mass accuracies of MS1 <10 ppm and MS2 <20 ppm, carbamidomethylation on cysteine and oxidation on methionine as variable modifications. The number of residues of each peptide on a crosslink pair was set between 4 and 100. A maximum of two trypsin missed-cleavage sites were allowed. The results were obtained with 1% false discovery rate. The identified crosslinks were filtered with a threshold of at least two spectral counts and a pLink score < 10e-4.

## Acknowledgements

We would like to thank Ulrike Teichmann and Rolf Rümenapf for animal care and immunization; Jens Krull, Renate Rees, Heinz-Jürgen Dehne and Gabriele Hawlitscheck for excellent technical assistance; Volker Cordes for sharing the HeLa GFP-Nup153 cell line; Koray Kirli for help with sequence analysis; Alpacas Olga and Doris for kindly sharing their exquisite immune repertoire; the crystallization facility of our institute for providing the robotic screening infrastructure, Nataliia Naumenko and Cornelia Paz for critical reading of the manuscript and the Max-Planck-Gesellschaft as well as the DFG (SFB860) for funding this work. Coordinates and structure factor files have been submitted to the Protein Data Bank (PDB) and are available under the accession code 5E0Q.

## Additional information

### Funding

| Funder | Author |
|---|---|
| Deutsche Forschungsgemeinschaft | Holger Stark Henning Urlaub Dirk Görlich |

The funders had no role in study design, data collection and interpretation, or the decision to submit the work for publication.

## Author contributions
TP, Conception and design, Acquisition of data, Analysis and interpretation of data, Drafting or revising the article; MBa, ST, C-TL, JES, MBö, Acquisition of data, Analysis and interpretation of data; HC, Drafting or revising the article, Contributed unpublished essential data or reagents; HS, HU, Analysis and interpretation of data, Drafting or revising the article; DG, Conception and design, Analysis and interpretation of data, Drafting or revising the article

## Author ORCIDs
Tino Pleiner, http://orcid.org/0000-0002-5104-0315
Mark Bates, http://orcid.org/0000-0003-0668-5277
Sergei Trakhanov, http://orcid.org/0000-0002-1326-6153
Jan Erik Schliep, http://orcid.org/0000-0001-7103-5954
Hema Chug, http://orcid.org/0000-0002-9889-7076
Marc Böhning, http://orcid.org/0000-0002-4245-5113
Henning Urlaub, http://orcid.org/0000-0003-1837-5233
Dirk Görlich, http://orcid.org/0000-0002-4343-5210

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
