## [Decision Letter]

Thank you for submitting your work entitled "Nanobodies: site-specific labeling for superresolution imaging, rapid epitope-mapping & native protein complex isolation" for consideration by *eLife*. Your article has been reviewed by two peer reviewers, and the evaluation has been overseen by Karsten Weis as Reviewing Editor and Randy Schekman as the Senior Editor.

The reviewers have discussed the reviews with one another and the Reviewing Editor has drafted this decision to help you prepare a revised submission.

Summary:

There was consensus amongst the reviewers that the manuscript by Pleiner et al. is an impressive study that will be of great use to many labs in cell biology and biochemistry describing valuable tools and approaches for the specific study of the nuclear pore complex. The reviewers had no major scientific issues with the study but made multiple suggestions to edit the text. For example, the Abstract was considered to be not as strong or clear as the research deserves. Clarity could be achieved with the rewording of the Abstract to stronger emphasize the scientific achievements. Similar points were made for the main the text, too. See, for example, the fourth paragraph of the subsection “Site-specific fluorescent labeling of nanobodies”.

---

## [Author Response]

*There was consensus amongst the reviewers that the manuscript by Pleiner et al. is an impressive study that will be of great use to many labs in cell biology and biochemistry describing valuable tools and approaches for the specific study of the nuclear pore complex. The reviewers had no major scientific issues with the study but made multiple suggestions to edit the text. For example, the Abstract was considered to be not as strong or clear as the research deserves. Clarity could be achieved with the rewording of the Abstract to stronger emphasize the scientific achievements. Similar points were made for the main the text, too. See, for example, the fourth paragraph of the subsection “Site-specific fluorescent labeling of nanobodies”*.

We accommodated all suggested improvements to the Abstract and main text, added the requested experimental details, as well as the PDB accession code (5E0Q) for the anti-Nup98 nanobody TP377 - Nup98 crystal structure. The only suggestion we did not implement was to move Figure 5 to the supplements, because we still believe that this is a very central piece of data (showing that maleimide labeling of nanobodies works through at least 6 different ectopic cysteine positions).